# Do Progressive Sensorimotor Training Devices Produce A Graded Increase in Centre of Mass Displacement During Unipedal Balance Exercises in Athletes

**Nina Gras [1,\*], Torsten Brauner [2], Scott Wearing [1,3] and Thomas Horstmann [1,4]**

[1] Faculty of Sports and Health Sciences, Technische Universität München, 80992 Munich, Germany; s.wearing@qut.edu.au (S.W.); t.horstmann@tum.de (T.H.)

[2] Department of Sport Science, Germany University of Health & Sport, 85737 Ismaning, Germany; torsten.brauner@dhgs-hochschule.de

[3] School of Clinical Sciences & Institute of Health and Biomedical Innovation, Queensland University of Technology, Brisbane 4059, Australia

[4] Medical Park St. Hubertus, 83707 Bad Wiessee, Germany

\* Correspondence: nina.gras@tum.de

**Abstract:** Progression of the difficulty of agility exercises in sport is often achieved by changing the stability of the support surface via graded sensorimotor training devices. However, little is known about the challenge imposed to postural equilibrium by these graded devices. This study quantified the instability provided by four sensorimotor training devices typically used to enhance athletic performance; three progressively unstable balance pads (ST1–3) and an oscillatory platform (PM). Twenty-five (13 female, 12 male) young adults (age, 26 ± 3 yr; height, 1.76 ± 0.10 m; and weight, 69 ± 12 kg), completed seven unipedal balance conditions involving stable and progressively unstable surfaces that involved four sensorimotor training devices (ST1-3, PM) and their combination (PM-ST1, PM-ST2). An inertial sensor, mounted over the lumbar spine, was used to monitor Centre of Mass (COM) displacement in each condition. Potential differences in COM displacement between conditions were assessed using a mixed-model analysis of variance. COM displacement differed between training devices; with a progressive, though non-linear, increase in COM displacement from the most (ST1) to the least (ST3) stable balance pad. However, there was no significant difference in COM displacement between the least stable balance pad (ST3) and the oscillatory platform used in isolation (PM) or in combination with balance pads (PM-ST1, PM-ST2). These novel findings have important practical implications for the design of progressive sensorimotor training programs in sport.

**Keywords:** postural equilibrium; balance training; instability devices; inertial sensor

## 1. Introduction

Neuromuscular or proprioceptive training interventions are commonly used in elite sports to enhance athletic performance, reduce the risk of certain types of sports injuries and hasten recovery post injury [1–3]. It is now widely accepted that the neuromuscular system adapts specifically to the applied training dose [4–6], and that the overall training stimulus in sensorimotor training sessions should be gradually increased over time [7]. One approach to increasing the training stimulus is to increase the duration of neuromuscular exercises or the number of sets. However, comparable improvements in balance performance have been observed in athletes irrespective of whether the

progression of balance exercises are time- or repetition-based [8]. An alternative approach to increasing the training stimulus is to increase the intensity of the exercise. The American College of Sports Medicine currently recommends that the intensity of neuromuscular exercise may be increased by reducing the base of support (e.g., from bipedal to unipedal stance) or manipulating the stability of the support surface. In the sports performance and rehabilitation setting, this is often achieved through using an ever-growing number of instability devices [9]. Although sensorimotor exercises performed on an unstable surface have generally been shown to promote synergistic neuromuscular activity [10–12], to date, most studies evaluating the effect of instability on neuromuscular training have focused on aspects of muscular strength [13–15], rather than proprioception and balance per se [16]. Moreover, in a systematic review of sensorimotor training programs, Lesinski et al. [17] identified that optimal training parameters for steady-state balance improvement included a training period of 11–12 weeks, involving 2 sets of 4 exercises, each with a duration of 20–40 s per exercise, undertaken at a frequency of 3 to 6 times per week. The authors were unable to draw conclusions regarding the optimal intensity of exercise, however, citing a lack of detailed research on the topic as a major impediment [17]. Thus, when planning the composition of sensorimotor training programs for improved steady-state balance, coaches and sports trainers lack reference data for the selection of the most appropriate instability device from the wide range of existing balance equipment.

This study aimed to quantify the challenge to postural equilibrium provided by four different instability training devices that are typically used to improve sensorimotor control in athletes; three balance pads of progressive stiffness and an oscillatory platform [12,18]. As body-worn accelerometers positioned over the lumbar spine allow field-based measures and have been shown to accurately assess the movement of the body's center of mass (COM) during balance training [19,20], this approach was used to compare the level of instability provided by each balance device. It was hypothesized that: (1) progressively unstable balance pads would present a progressively greater challenge to balance and, therefore, an increase in COM displacement; (2) the free translatory movement of an oscillating platform would produce greater instability and induce greater COM displacement than instability pads; and (3) the combination of an oscillating platform and instability pads would represent the most unstable balance condition and result in greater COM displacement than either device used in isolation. Finally, given that previous studies in older adults have identified differential effects of sex on static balance performance [21–23], we hypothesized that males would present with greater COM displacement than females during each balance condition.

## 2. Materials and Methods

This study used a repeated measures design, in which each participant completed a reference condition (bilateral barefoot stance) and seven balance conditions involving unipedal stance on stable (floor) and unstable surfaces (instability training devices).

### 2.1. Participants

Twenty-five healthy young athletes, aged between 18 and 35 years, were recruited from the Department of Sport and Health Sciences at the Technical University of Munich. The mean age, height and body mass of participants is shown in Table 1. All participants regularly participated in structured field sports (1 to 4 h per week) and were free from injury involving the spine and lower extremities. No participant reported a history of medications use or medical conditions known to affect balance. Participant numbers were based on unpublished pilot data for healthy adults and were sufficient to detect a 15% difference in normalized COM displacement ($\alpha = 0.05$, $\beta = 0.20$). All participants provided written informed consent to the procedures of the study, which received approval from the Institutional Ethics Committee.

**Table 1.** Mean (SD) participant characteristics.

|  | Female | Male | Average |
|---|---|---|---|
| n | 13 | 12 | 25 |
| Age (years) | 26 | 28 | 26 |
|  | (3) | (2) | (3) |
| Height (m) | 1.68 | 1.84 * | 1.76 |
|  | (0.06) | (0.05) | (0.10) |
| Body Mass (Kg) | 61 | 78 * | 69 |
|  | (7) | (9) | (12) |
| Body Mass Index (Kg/m$^2$) | 21.7 | 22.9 * | 22.3 |
|  | (2.1) | (1.9) | (2.1) |

* Indicates a statistically significant difference ($p < 0.05$)

### 2.2. Balance Devices

Four different instability training devices typically used in sensorimotor training of athletes were evaluated (Figure 1), three Balance Pads (TheraBand® Stability Trainers, The Hygenic Corporation, Akron, USA) and an oscillatory platform (PM, Posturomed, Haider Bioswing GmbH, Pullenreuth, Germany) [12,18]. According to the manufacturer, the stability training pads provide a progressive level of instability from easy (ST1) through intermediate (ST2) to difficult (ST3). ST1 and ST 2 were comprised of PVC foam of increasing compliance, while ST3 consisted of a deformable air-filled rubber chamber (Figure 1). The PM consisted of a rigid platform (60 × 60 cm) suspended by four springs within a frame that allowed damped translation of 80 mm in the anteroposterior direction and 50 mm in the mediolateral translation with an oscillation frequency between 1.0 and 3.2 Hz.

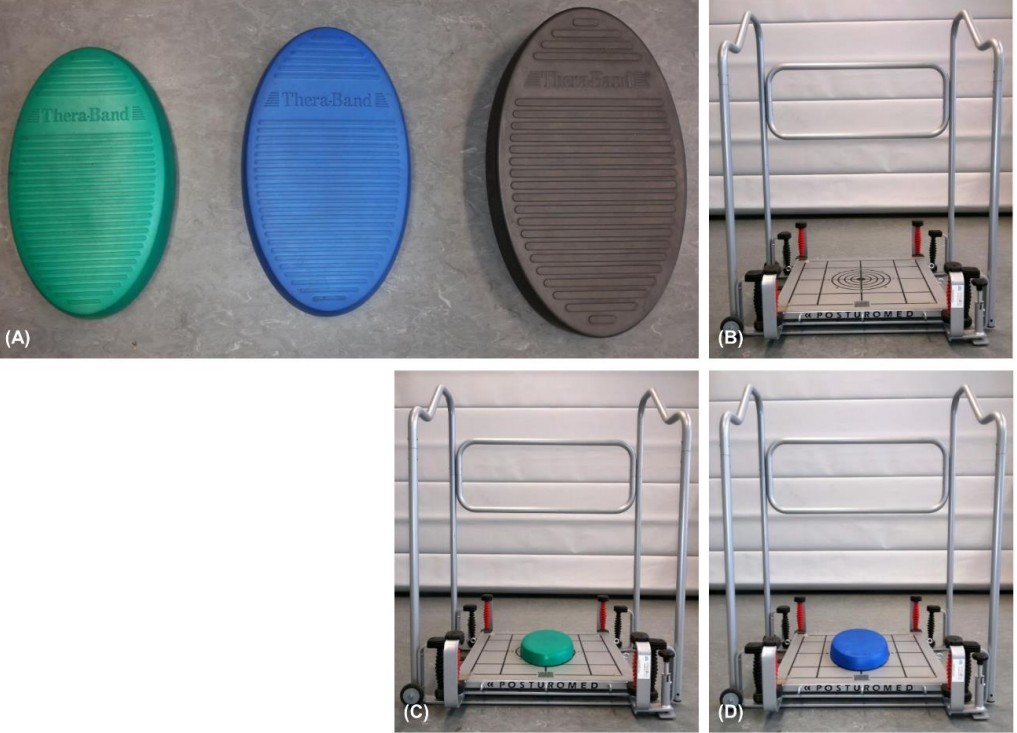

**Figure 1.** Illustration of the different instability devices used in sensorimotor balance conditions. Unipedal balance conditions requiring an upright stance were undertaken on a hard, flat surface (not shown), on three balance pads that, according to the manufacturer, progressed in difficulty from easy (ST1), through intermediate, (ST2) to difficult (ST3) (**A**), an oscillatory platform (PM) that allowed anteroposterior and mediolateral movement (**B**), and the platform combined with the most stable balance pad (PM-ST1) (**C**), and with an intermediate balance pad (PM-ST2) (**D**).

### 2.3. Protocol

Participants reported for sensorimotor measures wearing lightweight, comfortable clothing. In accordance with previous research [19,20], a triaxial inertial sensor unit (Data logger MSR160®, MSR Electronics GmbH, Seuzach, Switzerland) was positioned over the lumbar spine of each participant during erect upright stance and firmly secured via Velcro straps (Figure 2). The inertial sensor unit was $39 \times 23 \times 72$ mm in dimension and weighed 69 g. The full-scale of the sensor unit was $\pm$ 15 g, with a measurement resolution of 1%. The sensor unit was externally triggered and recorded to PC via a cable-mounted signal.

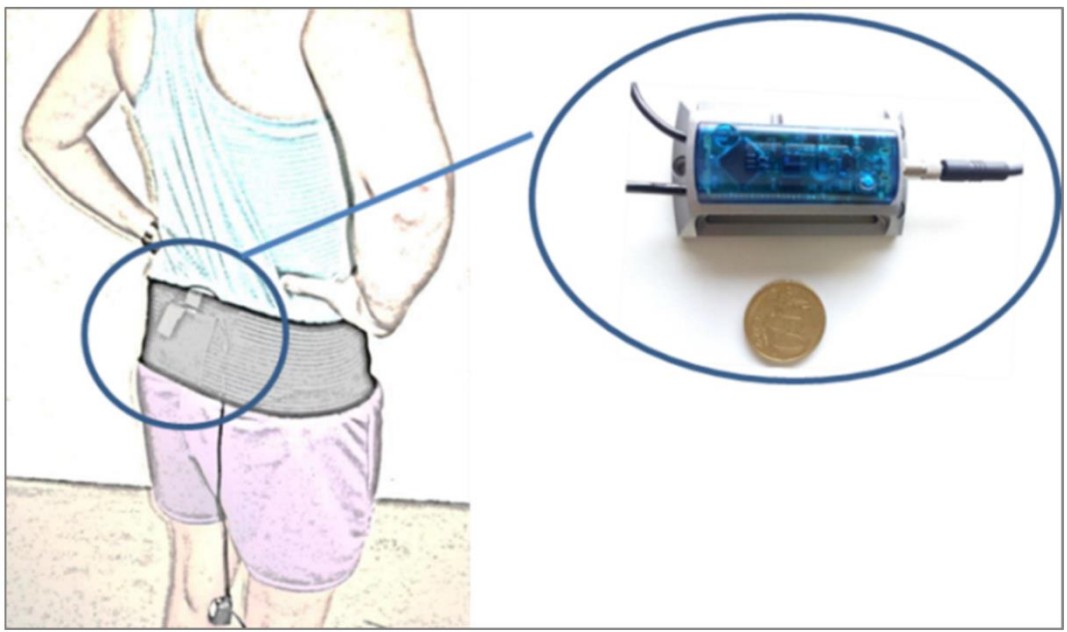

**Figure 2.** Positioning of the inertial sensor unit.

After a five-minute familiarization period, which included practice on each device, participants were evaluated during quiet bipedal stance on a stable surface (reference condition) and during seven balance conditions involving unipedal stance. Unipedal balance conditions included erect upright barefoot stance on; (1) a hard stable surface (floor); (2) the most stable balance pad (ST1); (3) the intermediate balance pad (ST2); (4) the most unstable balance pad (ST3); (5) the oscillatory platform (PM); (6) the oscillatory platform combined with the most stable balance pad (PM-ST1); and (7) the oscillatory platform combined with the intermediate balance pad (PM-ST2).

For each unshod balance condition, participants were instructed to stand as still as possible with hands on hips, eyes open, and looking directly forward. Unipedal stance conditions were undertaken for the right and left limb. The order of each balance condition was randomized between participants. The duration of each balance condition was 23 s, during which acceleration data were collected at a sampling rate of 1600 Hz. Two trials were collected for each condition, with a 10 s break provided between trials.

### 2.4. Data Processing and Statistical Analysis

Acceleration data were analyzed using custom computer code (MATLAB R2011b, MathWorks, Natick, MA, USA). The initial 3 s from each trial were excluded, to minimize potential stabilization effects that may have occurred with commencement of the exercise. Displacement of the COM was then calculated by double integration of the horizontal component of the acceleration data. Data were then high-pass filtered (4th order Butterworth, cut off frequency 0.5 Hz) to remove low frequency drift. The total path length of the COM was then calculated by summing the displacement values. For each

participant, the total path length of the COM for each balance condition was divided by that of the baseline condition (bipedal stance) and expressed as a proportion.

The statistical package R (version 3.0.3, R Foundation for Statistical Computing, Vienna, Austria) was used for all statistical procedures. Of the 400 trials, there were 12 trials in which participants were unable to maintain balance for the full test duration (3% of observations). Hence, an intention-to-treat analysis was used in which the "worst case carried forward" method was employed for imputation of missing values.

Potential differences in COM displacement between varying sensorimotor training devices were assessed using a mixed-model analysis of variance (ANOVAs) within a generalized linear modeling framework. Balance condition (Floor, ST1, ST2, PM, ST3, PM-ST1, PM-ST2) was treated as a within-subject factor, while sex (male and female) was treated as a between-subject factor. Underlying assumptions regarding the uniformity of the variance–covariance matrix were assessed using Mauchly's test of sphericity. When the assumption of uniformity was violated, an adjustment to the degrees of freedom of the F–ratio was made using Greenhouse–Geisser Epsilon, thereby making the F–test more conservative. Post-hoc analysis consisted of paired *t*-tests. Effect sizes for differences in COM displacement between balance conditions were estimated using Cohen's d for repeated measures (Dz), in which the mean difference between balance conditions was standardized by the deviation of the difference [24]. As a general guideline, Dz in the range of 0.20–0.50 was considered to be a small effect, 0.50–0.80 a moderate effect, and > 0.80 a large effect [24].

## 3. Results

COM displacement during each balance condition for males and females is shown in Figure 3. There was no statistically significant main effect of sex on normalized COM displacement ($F_{1,23} = 2.7$, $p = 0.12$). Similarly, there was no statistically significant interaction between sex and balance condition ($F_{10,230} = 1.3, p = 0.25$). There was, however, a significant main effect of balance condition on normalized COM displacement ($F_{10,230} = 40.7$, $p < 0.001$). Post hoc analysis identified significant differences between unipedal stance conditions (Table 2). As shown in Figure 3, mean COM displacement increased progressively across the three balance pads ($p < 0.05$), with the least stable pad (ST3) invoking almost three (2.9) times greater COM displacement than the floor. There was, however, no statistically significant difference in mean COM displacement during unipedal stance on the least stable balance pad (ST3) and the oscillatory platform when used either in isolation (PM) or in combination with a balance pad (PM-ST1, PM-ST2). Although mean displacement of the COM on the oscillatory platform (PM) was approximately twice (2.3) that of unipedal stance on the floor ($p < 0.05$), it was not significantly different from that of the intermediate balance pad (ST2) or when it was used in combination with the most stable balance pad (PM-ST1).

**Table 2.** *P*-values and effect sizes (*dz*) for post hoc comparisons between balance conditions.

|  | Floor | ST1 | ST2 | ST3 | PM | PM-ST1 |
|---|---|---|---|---|---|---|
| ST1 | 0.02 | - | - | - | - | - |
| *dz* | 0.82 | | | | | |
| ST2 | 0.00 | 0.00 | - | - | - | - |
| *dz* | 1.54 | 1.13 | | | | |
| ST3 | 0.00 | 0.00 | 0.00 | - | - | - |
| *dz* | 1.99 | 1.61 | 1.30 | | | |
| PM | 0.00 | 0.04 | 1.00 | 1.00 | - | - |
| *dz* | 1.02 | 0.44 | 0.47 | 0.45 | | |
| PM-ST1 | 0.00 | 0.01 | 0.06 | 1.00 | 1.00 | - |
| *dz* | 0.98 | 0.93 | 0.74 | 0.09 | 0.44 | |
| PM-ST2 | 0.00 | 0.00 | 0.00 | 1.00 | 0.02 | 1.00 |
| *dz* | 1.53 | 1.09 | 3.81 | 0.31 | 0.81 | 0.35 |

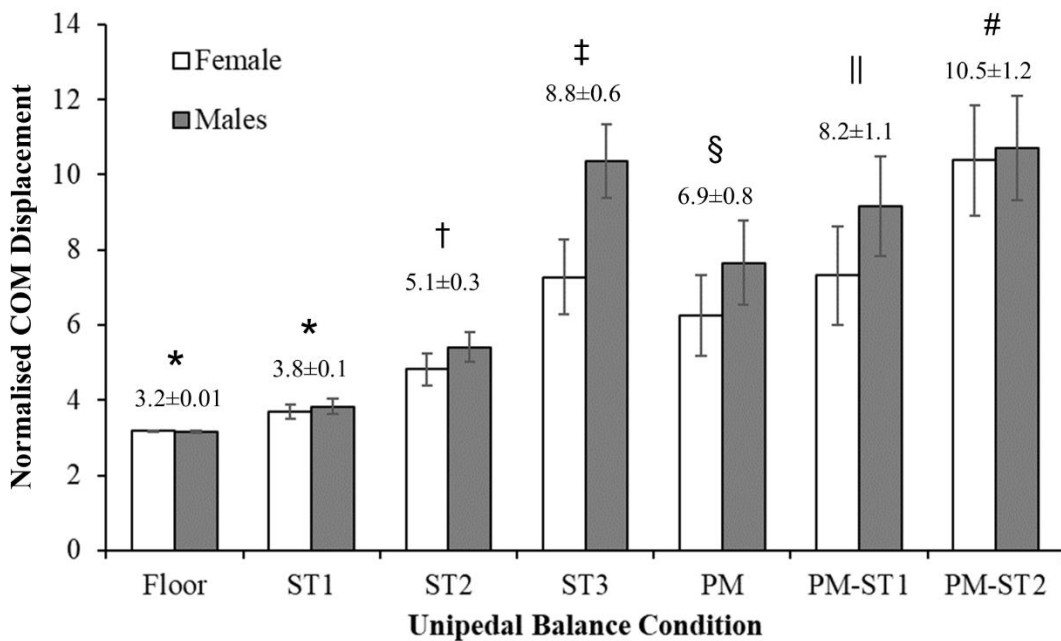

**Figure 3.** Mean normalized center of mass (COM) displacement for all balance conditions in females and males. Error bars represent standard error. Note that there were no significant differences between males and females. Mean (± standard error) normalized COM displacement values are shown for each balance condition. * Balance condition significantly different from all other conditions ($p < 0.05$). † Balance condition significantly different from all other conditions except PM and PM-ST1 ($p < 0.05$). ‡ Balance condition significantly different from all other conditions except PM, PM-ST1 and PM-ST2 ($p < 0.05$). § Balance condition significantly different from all other conditions except PM-ST1 ($p < 0.05$). ‖ Balance condition significantly different from all other conditions except ST2, ST3 and PM-ST2 ($p < 0.05$). # Balance condition significantly different from all other conditions except ST3 and PM-ST1 ($p < 0.05$).

COM displacement during stance on the most stable balance pad (ST1) was significantly correlated with that on the intermediate (ST2) balance pad (r = 0.79, $p < 0.001$). Similarly, COM displacement during stance on the oscillatory platform combined with the most stable balance pad (PM-ST1) was moderately correlated with that of its combined use with intermediate (PM-ST2) balance pad (r = 0.71, $p = 0.001$). There were no other significant correlations between balance conditions.

## 4. Discussion

It is well known that introducing instability during static balance increases sensory input, promotes central nervous system processing, as well as motor actions and reactions to moderate motor commands. Neurophysiological studies have demonstrated central adaptations at multiple levels due to exercises using unstable surfaces [25], including increased co-activation of agonist and antagonist muscles that are related to the level of instability [12,26,27]. Although coaches and sports trainers generally agree on qualitative ratings of instability devices, objective quantification of the grade of instability and the challenge imposed on postural equilibrium by these devices is currently lacking. The primary purpose of this study, therefore, was to quantify the amount of COM displacement induced by different balance devices commonly used in sensorimotor training in a young athletic population. Consistent with our hypothesis and the findings of Wolburg et al. [12], mean COM displacement during unipedal stance sequentially increased with progressively unstable balance pads. While unipedal stance alone induced approximately three-fold greater COM displacement than the referent bipedal stance condition, instability balance pads resulted in a further exponential increase in COM excursion, from approximately four times that of bipedal stance with the most stable pad, to nine times that of bipedal stance with the least stable pad. Interestingly, although balance performance on the two

most stable pads (ST1 and 2) was moderately correlated, performance on the least stable, air-filled balance pad (ST3) was unrelated to performance on either of the foam pads (ST1 and 2). Hence, while instability pads appear to provide a progressive stimulus for sensorimotor coordination in healthy young athletes, coaches and sports trainers should be aware that the stimulus across the three graded balance pads is not linear.

It was also hypothesized that an oscillating platform would induce greater COM displacement than instability pads. In partial support of this hypothesis, significantly greater COM displacement was observed during stance on the oscillating platform (PM) than on the most stable balance pad (ST1). However, contrary to our hypothesis, no statistically significant difference in COM displacement was observed between the oscillating platform and the intermediate (ST2) or the least stable balance pad (ST3). Hence, the oscillating platform provided no greater perturbation to balance in healthy young adults, on average, than either the intermediate or the least stable balance pad (ST2 and ST3); even when used in combination with ST1. The variability between participants, however, was greatest for the oscillatory platform when used either in isolation (PM) or in combination with balance pads (PM-ST1, PM-ST2). The between-subject variance in COM displacement was approximately seven times greater during stance on the oscillatory platform (PM) than on a stable hard, flat surface (Floor) and was approximately 12-fold greater during combined conditions (PM-ST1, PM-ST2). Hence, the challenge to sensorimotor control induced by the oscillating platform differed between participants. Although the current study did not include measures of lower limb joint moments, one possible explanation for the greater between-subject variability is that the oscillatory platform may provoke a different balance strategy than balance pads. For instance, Runge et al. [28] showed that rapid translations on flat surfaces, similar to that imposed by the oscillatory platform, result in a shift of postural control strategies from either an ankle or hip strategy [29], to a more combined balance strategy. Similarly, Wolburg et al. [12] reported that hard surfaces provoke a more rigid ankle joint and a shift towards a hip strategy. With this in mind, it is noteworthy that performance on balance conditions involving two of the foam-based balance pads (ST1 and ST2) remained moderately correlated, even when used in combination with the oscillating platform. Hence, combined use of foam balance pads with an oscillatory platform likely invokes an interplay between different postural control strategies. Although further research is required, it is possible that the least stable balance pad and oscillatory platform may be a useful tool for training the adaptation of balance strategies and multi-joint coordination in young athletes, or for identifying athletes who are weak strategy adapters.

Although previous studies have identified differential sex effects in static postural control in older adults [21–23], there were no significant differences in COM displacement between males and females during any balance conditions in this study. The finding is consistent with studies of postural sway in healthy young adult populations [21–23], and collectively, suggest that sex-related differences in balance may be more prevalent in elderly adult populations. Given the difference in height and weight of males and females in the current study, our results also lend indirect support to previous research in which measures of body anthropometry, such as stretch stature and weight have been shown to be only weakly correlated to static postural control in healthy adults [30,31].

This study has several limitations that should be kept in mind when interpreting the results. First, although participants in the current study were afforded a familiarization period on each device [32] and the order of balance conditions was randomized between participants to minimize potential learning effects, training-related enhancements in postural control and muscular activity are well documented in balance-related tasks. For instance, Brueckner et al. [33] showed that as little as two days of training on a wobble board induced adaptive responses in the neuromuscular system that enhanced postural control and balance. The relative challenge imposed by each sensorimotor device to postural equilibrium in the current study, therefore, should be considered as a baseline response only, which is likely to change with extended exposure. Secondly, this study quantified COM displacement using a body-worn triaxial accelerometer rather than using a force platform to monitor center of pressure movement as traditionally used in static posturography. Although accelerometers have been

shown to be reliable, accurate, and sensitive measures of postural control [19,20,34], the combined use of these and other technologies, such as 3D motion analysis, would have undoubtedly provided greater insight into potential joint loading and balance strategies employed by participants on each device [35]. Further research incorporating motion analysis, ground reaction force, and electromyography, therefore, is required to evaluate potential neurological mechanisms and balance strategies employed by participants on each device. Finally, this study evaluated COM displacement in healthy young athletes competing in field-based sports. Given that the control of posture involves different physiological systems that can be affected by training, pathology, or sub-clinical constraints [25], the results of the current study may not be directly transferable to other athletic cohorts, older adults, or patient groups. Nonetheless the current study provides coaches and sports trainers with a quantitative index of the postural challenge imposed by common instability devices to aid in the design of appropriate progressive sensorimotor training programs in young adults.

## 5. Conclusions

The findings of the current study show that progressively unstable balance pads result in a non-linear increase COM displacement during unipedal stance in healthy young athletes. The use of an oscillating platform, in isolation or combination, provided minimal additional increase in COM displacement, on average, than the most unstable balance pad. These novel results provide quantitative information concerning the challenge imposed by progressive balance conditions and serve as a guide for sports trainers and athletic coaches who wish to design progressive sensorimotor balance exercises for young athletes.

**Author Contributions:** Conceptualization, N.G., T.B. and T.H.; methodology, N.G., T.B. and T.H.; formal analysis, N.G. and S.W.; investigation, N.G.; resources, T.B. and T.H.; data curation, N.G., and T.B.; writing—original draft preparation, N.G. and S.W.; writing—review and editing, N.G., T.B., S.W. and T.H.; visualization, N.G., T.B. and T.H.; supervision, T.H.; project administration, T.B. All authors have read and agreed to the published version of the manuscript.

**Funding:** This research received no external funding. The TheraBand® Stability Trainers used in this study were donated by Ludwig Artzt GmbH.

**Conflicts of Interest:** The authors declare no conflicts of interest.

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
