# Peer review of "Do Progressive Sensorimotor Training Devices Produce A Graded Increase in Centre of Mass Displacement During Unipedal Balance Exercises in Athletes"

_applsci, doi:10.3390/app10113893_

Round 1
Reviewer 1 Report
Introduction
Hypotheses were formulated in the introduction, explaining what was expected in the study, but no explanation on why do authors expected the differences between each conditions was included
Materials and methods
Twenty-five healthy young athletes were recruited to participate in the study. The groups were relatively small, plus the authors didn’t mention what kind of sport participants participated in? Maybe that influenced the results?
The authors proposed several balance conditions. Why oscillatory platform was used only in combination with balance pads ST1, ST2 (PM-ST1, PM-ST2)?
Results - discussion
In general, the own results are not discussed in detail
For instance, the aim of the study was to compare COM displacement in several balance conditions. However, the different results of proposed conditions are not discussed very well. What might be the explanation for the observed changes?
Also authors wrote about gender differences in COM displacements in results/discussion section but there is no information or hypothesis included in the introduction section. The lack of differences in COM displacement between males and females during balance conditions was not discussed - line 228. Does the difference in height between males and females may influence balance performance?
Lines 224-226“Although further research is required, it is possible that the oscillatory platform may be a useful tool for training the adaptation of balance strategies and multi-joint coordination in young athletes or for identifying athletes that are weak strategy adapters”- On what grounds do authors indicate that?

Author Response
REVIEWER 1
Introduction
- Hypotheses were formulated in the introduction, explaining what was expected in the study, but no explanation on why do authors expected the differences between each conditions was included.
We have now provided a rationale for each hypothesis. In response to the reviewer’s subsequent comment (comment 5), we have now also included an hypothesis related to potential sex-effects on balance performance along with an appropriate rationale.
[Lines 69-76]
Materials and methods
- Twenty-five healthy young athletes were recruited to participate in the study. The groups were relatively small, plus the authors didn’t mention what kind of sport participants participated in? Maybe that influenced the results?
As outlined in the original manuscript, the sample size was based on pilot data for healthy adults and was sufficient to detect a 15% difference in normalized COM displacement (α=.05, β=.20). The athletes involved in this study participated in a variety of field-based sports. We have now modified the methods to include this information. While we appreciate the reviewer’s comment, we believe it is not clear how sports participation may influence balance performance. For instance, in a systematic review of athletes from a variety of sports, Zemková (2014) concluded there were no significant differences in postural stability between athletes from different specialties. Nonetheless, given the reviewer’s comment, we have now modified the discussion to indicate that the study involved young athletes from field-based sports and, given that control of posture involves different physiological systems that can be affected by training or pathology, the results may not be directly transferable to other athletic cohorts, older adults or patient groups.
[Lines 267-71]
Reference
Zemková, E. Sport-specific balance. Sports Med. 2014;44(5):579-90.
- The authors proposed several balance conditions. Why oscillatory platform was used only in combination with balance pads ST1, ST2 (PM-ST1, PM-ST2)?
We had originally planned to undertake all possible balance combinations, however, we did not evaluate the combination of the least stable balance pad and the oscillatory platform due to potential health and safety concerns.
Results - discussion
- In general, the own results are not discussed in detail. For instance, the aim of the study was to compare COM displacement in several balance conditions. However, the different results of proposed conditions are not discussed very well. What might be the explanation for the observed changes?
It is well known that instability during static balance induces a degree of uncertainty and, therefore, increases sensory input, CNS processing, and motor actions and reactions to adjust motor commands (Taube et al., 2008, Franklin and Wolpert, 2011). Neurophysiological studies have demonstrated central adaptations at multiple levels due to exercises using unstable surfaces, including increased cerebellar and subcortical activity in combination with reduced spinal reflex excitability and cortical activity, and that these adaptations are task specific (Taube et al., 2008). A common finding during initial unstable task training is increased co-activation of agonist and antagonist muscles (Burdet et al., 2001, Franklin et al., 2003), related to the level of instability (Franklin et al., 2004, Selen et al., 2009). Although coaches and sports trainers generally agree on qualitative ratings of instability devices, objective quantification of the grade of instability and the challenge imposed on postural equilibrium by these devices is currently lacking. The primary purpose of this study, therefore, was to quantify the COM displacement induced by different balance devices and their combination. We hypothesized that progressively unstable balance pads should provide a progressive challenge to balance performance, as marketed, and subsequently a linear increase in COM displacement. Such reference data would assist coaches and sports trainers in the selection of the most appropriate instability condition and to plan the composition and timing of sensorimotor training programs.
In light of the reviewer’s comment, we have now further emphasized the aim of the study. Although the study was not designed to be mechanistic, we have now speculated to potent mechanisms behind the increased COM displacement within the discussion.
[Lines 197-204]
References
Burdet, et al. Nature 2001;414 (6862):446-449.
Franklin et al. J Neurophysiol, 2003;90(5):3270-3282.
Franklin, et al. J Neurophysiol, 2004;92:3097-3105.
Franklin, D.W. Wolpert, D.M. Neuron, 2011;72 (3):425-442.
Selen, et al. J Neurosci, 2009;29(40):12606-12616.
Taube, et al. Acta Physiol. 2008; 193 (2):101-116
- Also authors wrote about gender differences in COM displacements in results/discussion section but there is no information or hypothesis included in the introduction section. The lack of differences in COM displacement between males and females during balance conditions was not discussed - line 228. Does the difference in height between males and females may influence balance performance?
In light of the reviewer’s comment, we have now provided the rationale for comparing the balance performance of males and females. We have also included a specific hypothesis based on the literature, that males would have greater COM displacement than females during all balance conditions.
[Line 74-6]
We highlighted within the discussion of the original text that, contrary to previous literature involving elderly adults, we observed no significant significant sex effects in static balance performance in our young athletic cohort. Rather our results are consistent with research that has evaluated sex-related differences in healthy young adults. Thus, although beyond the scope of the present research, it would appear that sex-related differences in balance may be more prevalent in elderly adult populations. We also concur with the reviewer that some studies have suggested increased body height may negatively influence balance performance. However, the weight of literature indicates that body height is only weakly correlated to COM displacement during static balance in healthy adults (Kejonen et al 2003; Alonso et al 2012). Although not an aim of this study, our results would tend to support these findings. At the request of the reviewer, we have now broadened the discussion to speculate the potential impact of body anthropometry on static postural control.
[Line 246-50]
References
Kejonen et al. Arch Phys Med Rehabil. 2003;84(1):17–22.
Alonso et al. Clinics 2012;67(12):1433‐1441.
- Lines 224-226“Although further research is required, it is possible that the oscillatory platform may be a useful tool for training the adaptation of balance strategies and multi-joint coordination in young athletes or for identifying athletes that are weak strategy adapters”- On what grounds do authors indicate that?
As highlighted in the discussion [lines 211-26], we observed that performance on balance conditions involving two of the foam-based balance pads (ST1 and ST2) remained moderately correlated, even when used in combination with the oscillating platform. However, the between-subject variance in COM displacement was markedly higher in conditions involving the oscillatory platform. We proposed that one possible explanation for the greater between-subject variability is that the oscillatory platform may provoke a different balance strategy than balance pads [Lines 219-21] and highlighted previous research that supports this concept (eg Runge et al.[22] [23] and Wolburg et al.[12]). On this basis we believe it is appropriate to argue that incorporation of the oscillatory platform may be a useful tool for training the adaptation of balance strategies or for identifying athletes that are weak strategy adapters within the conclusion. In light of the reviewer’s comment, we have now further emphasized this point within the text.
[Line 238-9]
Reviewer 2 Report
The authors aimed to quantify the challenge to postural equilibrium provided by four different instability training devices that are used to improve sensorimotor control in athletes.
The manuscript is well written.
Minor comments:
- line 140 - please specify how was the COM normalized to the baseline condition? The unit of measurement?
- Could the authors provide the mean values for COM displacement for each situation (in a table)?
- Line 185-186 - is the COP or COM displacement?
Author Response
REVIEWER 2
The authors aimed to quantify the challenge to postural equilibrium provided by four different instability training devices that are used to improve sensorimotor control in athletes.
- The manuscript is well written.
We thank the reviewer for the kind comment
Minor comments:
- line 140 - please specify how was the COM normalized to the baseline condition? The unit of measurement?
For each participant, the total path length of the COM for each balance condition was normalised by dividing it by the total path length of the COM during the baseline condition (bipedal stance) and was subsequently expressed as a proportion. We have now updated the text accordingly.
[Lines 144-5]
- Could the authors provide the mean values for COM displacement for each situation (in a table)?
In order to address this concern and minimize the number of figures and tables, we have now included mean values for each balance condition within Figure 3.
[Figure 3]
- Line 185-186 - is the COP or COM displacement?
We thank the reviewer for identifying this oversight and have now changed the term COP to COM.
[Lines 189, 190, and 225]
Reviewer 3 Report
Please describe what is new at work in the summary. Are there currently any alternatives for this device on the market? It would be good to include load schemes for each human joint while working on the platform. Please complete and current literature in the introduction.
Author Response
REVIEWER 3
- Please describe what is new at work in the summary. Are there currently any alternatives for this device on the market?
Progression of the difficulty of balance exercise in sport is commonly achieved by changing the stability of the support surface via graded sensorimotor training devices. However, little is known about the challenge imposed to postural equilibrium by these devices. This study quantified, for the first time, the challenge imposed to postural equilibrium by graded sensorimotor training devices. It specifically considered commonly used devices from many available on the market. Given, the reviewers comment, we have now provided a more explicit rationale for the research within the abstract and discussion and have also emphasised the novel findings within the conclusion
[Lines 200-8, 215-9, 279-83]
- It would be good to include load schemes for each human joint while working on the platform. Please complete and current literature in the introduction.
This study used body-worn accelerometers to assess COM displacement during single-limb stance on balance devices that are commonly used in sport. Although accelerometers allow accurate field-based measures of COM movement, they do not allow for estimates of joint loading. While we agree with the reviewer that additional information, such as lower limb kinematics, kinetics and joint moments, may provide greater insight into potential strategies used by athletes during the different balance conditions, we limited the aim of the study to an evaluation of the relative challenge imposed by each sensorimotor device to postural equilibrium. We believe that this is a necessary first step in evaluating the utility of such devices. Nonetheless, given the reviewer’s comment, we have now modified the discussion to further emphasise that this is an area for further research.
[Line 263-4]
Reviewer 4 Report
The abstract should not contain details of statistics (like p values or alpha), which must be reported in the results section. Please, report the full name of PM the firs time you mention it (i.e. in abstract section).
In the introduction, authors should describe the rationale for comparing males vs female performance, and indicate whether they have a specific hypothesis about gender differences.
My main concern is about research design: 1) it is not clear whether subjects had already experience with the devices, which could be quite familiar for athletes. If so, I expect a bias in the data set, which should be taken into account. Furthermore, 2) I expect that subjects can learn to balance on each device, as authors declare in the discussion section. Although a 5-minute familiarization period has been used here, I guess that differences emerged in the baseline score for each subject, as some of them can learn faster than others. This is confirmed by the evidence of a relevant variability among subjects. For these reasons, data for the familiarization period could be shown in the results section but more importantly, authors should normalize each subject's performance to his/her baseline performance. Actually, it is not clear if this has been done or not (I think that when using the term 'normalized COM displacement' authors refer to average values). 3) I don't see any rationale for not including PM-ST3 condition. Is that too difficult or potentially dangerous? Please specify
In methods section: 1) in Table 1 I would recommend to refer to average (rather than total) for all parameters except the number of participants. Also, 2) check average ages (as these do not correspond to the 18-35 years-range reported in the text). 3) why do authors refer to ST1 and ST3 vs 1-ST2) what the number 1 indicates? 4) how long each session lasted? data were collected for 23 seconds, is that the whole duration of each session? if not, how much time passed since the beginning of the session and the beginning of the data collection?
I have some concerns about results section: 1) Please correct COP with COM where necessary. 2) More importantly in figure 3, lines above males and females histograms for each condition are confounding: the symbols above these lines seem to indicate statistical differences between genders. 3) in figure 3 legend authors declare no gender differences, but it does not look like this for ST3 condition. Please check. 4) Statistical differences in figure3 are hard to understand. I would recommend to use group-lines rather than different symbols and the use of 'except for' that complicate the reading.
Conclusions: the evidence that "CPM displacement increases with balance pads" is intrinsic in the tools used (that is: it should be an independent variable), then at the end of the day the main result is that PM makes COM more complex, regardless of the balance pad used.
Author Response
REVIEWER 4
- The abstract should not contain details of statistics (like p values or alpha), which must be reported in the results section. Please, report the full name of PM the firs time you mention it (i.e. in abstract section).
We have now removed P values and the alpha level used for statistical analysis from the abstract in light of the request of the reviewer. Appropriate F statistics and P values reported within the results have been retained. We believe the reviewer may have overlooked that within the abstract we provided the abbreviation “PM” in parentheses immediately following the description of the device (ie oscillatory platform). Consequently, we have not modified the text.
[Lines 28-30, text deleted]
- In the introduction, authors should describe the rationale for comparing males vs female performance, and indicate whether they have a specific hypothesis about gender differences.
We have now provided the rationale for comparing static balance performance of males and females and included a specific hypothesis based on the literature, that males would have greater COM displacement than females under all balance conditions.
[Line 74-6]
- My main concern is about research design: 1) it is not clear whether subjects had already experience with the devices, which could be quite familiar for athletes. If so, I expect a bias in the data set, which should be taken into account. Furthermore, 2) I expect that subjects can learn to balance on each device, as authors declare in the discussion section. Although a 5-minute familiarization period has been used here, I guess that differences emerged in the baseline score for each subject, as some of them can learn faster than others. This is confirmed by the evidence of a relevant variability among subjects. For these reasons, data for the familiarization period could be shown in the results section but more importantly, authors should normalize each subject's performance to his/her baseline performance. Actually, it is not clear if this has been done or not (I think that when using the term 'normalized COM displacement' authors refer to average values). 3) I don't see any rationale for not including PM-ST3 condition. Is that too difficult or potentially dangerous? Please specify
Athletes were not undertaking agility training and reported no previous experience with the balance conditions evaluated in this study. As indicated in the original text [Line 129-30], the order of balance conditions was randomized between participants, hence negating the effect of potential learning effects. Further, each participant's performance was normalized to his/her baseline performance as stated in the original text [Line 144-5]. In light of the reviewer’s comment, we have now modified the text within the methods to further emphasis these points.
[Line 144-5, 255-7]
We agree with the reviewer that inclusion of the most unstable balance pad with the oscillatory platform would have resulted in a more balanced number of test conditions and had originally planned to undertake all possible combinations of stability pad and oscillatory platform. However, we did not evaluate the combination of the least stable balance pad with the oscillatory platform due to potential health and safety concerns. We believe, however, that the conditions that were evaluated still provided new insight into the postural challenge associated with their use.
- In methods section: 1) in Table 1 I would recommend to refer to average (rather than total) for all parameters except the number of participants. Also, 2) check average ages (as these do not correspond to the 18-35 years-range reported in the text). 3) why do authors refer to ST1 and ST3 vs 1-ST2) what the number 1 indicates? 4) how long each session lasted? Data were collected for 23 seconds, is that the whole duration of each session? If not, how much time passed since the beginning of the session and the beginning of the data collection?
- In accordance with the reviewer’s comment, we have now replaced the word “Total” in Table 1 with “Average.”
[Table 1, column title]
- We have now double checked the demographic data for men and women and corrected minor errors associated with rounding.
[Table 1]
- We thank the reviewer for identifying this oversight in naming convention. In reviewing the manuscript, we found three errors in naming convention and have subsequently corrected these.
[Lines 112, 128 and 226]
- The duration of each balance condition was 23 seconds. The text has now been modified accordingly
[Lines 134-5]
- I have some concerns about results section: 1) Please correct COP with COM where necessary. 2) More importantly in figure 3, lines above males and females histograms for each condition are confounding: the symbols above these lines seem to indicate statistical differences between genders. 3) in figure 3 legend authors declare no gender differences, but it does not look like this for ST3 condition. Please check. 4) Statistical differences in figure3 are hard to understand. I would recommend to use group-lines rather than different symbols and the use of 'except for' that complicate the reading.
- We thank the reviewer for identifying this and have now changed the term COP to COM. [Lines 189, 190, and 225]
- We appreciate the point raised by the reviewer. In considering the issue, we reconfigured Figure 3 to include group lines as suggested by the reviewer. However, in doing so, we found the nature of the differences and considerable overlap of lines was more confusing than use of symbols with each balance condition. Identifying the location of each difference, rather than citing the “exception” also resulted in the figure legend becoming text heavy and hard to follow. Consequently, in attempting to address the reviewer’s comment, we have now opted to retain the symbols but remove the parentheses from above each balance condition. We have also modified the figure legend to indicate that the symbol represents the balance condition and have also included the mean value for the balance condition at the request of reviewer 2.
[Figure 3]
- In light of the reviewer’s comment we have repeated the statistical analysis and can confirm that there is no statistically significant main effect for sex or a significant sex-condition interaction in the full factorial model. We can also confirm that cited F and P statistics in the results are correct. Consequently, we have not adjusted the text.
- Conclusions: the evidence that "CPM displacement increases with balance pads" is intrinsic in the tools used (that is: it should be an independent variable), then at the end of the day the main result is that PM makes COM more complex, regardless of the balance pad used.
The balance pads are marketed as providing a progressive or gradual increase in postural control. As highlighted within the original text, however, the balance pads resulted in an exponential increase in COM excursion. Hence, they provide a distinctly non-linear stimulus for sensorimotor coordination in healthy young athletes; such that the most unstable pad presented the same level of challenge as an oscillating platform, which clinically is thought to provide a greater stimulus. We have now modified the discussion to further emphasise this point.
[Lines 211-5, 238-40, 275-9]
Round 2
Reviewer 4 Report
Even if sex differences do not emerge through the factorial anova, graphs are quite evident, especially for ST3 group. I would recommend to double-check with Student t sex differences, and briefly discuss these results. Since the overall effect is not emerging, this request is not mandatory but sincerely reccomended in order to improve the manuscript.